# Betting, Selection, and Luck: A Long-Run Analysis of Repeated Betting Markets

**DOI:** 10.3390/e21060585

**Published:** 2019-06-13

**Authors:** Giulio Bottazzi, Daniele Giachini

**Affiliations:** Institute of Economics & Department EMbeDS, Scuola Superiore Sant’Anna, Piazza Martiri della Libertà 33, 56127 Pisa, Italy

**Keywords:** bounded rationality, betting strategies, market selection, recurrent processes, luck

## Abstract

We consider a repeated betting market populated by two agents who wage on a binary event according to generic betting strategies. We derive new simple criteria, based on the difference of relative entropies, to establish the relative wealth of the two agents in the long-run. Little information about agents’ behavior is needed to apply the criteria: it is sufficient to know the odds traders believe fair and how much they would bet when the odds are equal to the ones the other agent believes fair. Using our criteria, we show that for a large class of betting strategies, it is generically possible that the ultimate winner is only decided by luck. As an example, we apply our conditions to the case of Constant Relative Risk Averse (CRRA) and quantal response betting.

## 1. Introduction

Consider a repeated market for betting where two agents wage on the outcomes of a binary event. Agent behavior is described by generic betting strategies that depend on prevailing odds. If the odds are fixed, the strategy that guarantees optimal wealth growth is the Kelly rule [1,2]. If the odds depend on agents’ bets via the parimutuel procedure, in a population of Kelly bettors, the one with the most accurate beliefs accrues all the wealth and asymptotically dominates the market [3]. This is a particular case of the results derived in the equivalent setting of inter-temporal general equilibrium models with short-lived securities [4,5]. In a similar framework, an agent adopting the Kelly strategy and having correct beliefs will surely retain a positive wealth share, that is he/she survives, when trading against price-independent rules [6]. The global behavior of the Kelly strategy with perfect information being established, it remains to understand what happens when agents do not bet according to Kelly and/or are not perfectly informed. In a market populated by utility maximizers, the agent who trades knowing the correct probabilities always realizes a non-negative expected profit [7]. In the case of bettors using the fractional Kelly rule, a generalization of the Kelly rule that includes a risk-aversion parameter, sufficient and, apart from hairline cases, necessary conditions for strategy dominance or survival has been derived [8,9]. These conditions generalize and correct previous tentative results based on numerical simulations [10]. In intertemporal equilibrium models, when more general strategies are adopted, one can observe path-dependent cases, in which the agent who dominates depends on the sequence of realized events [11]. In general equilibrium models, a necessary and sufficient condition for a trader to vanish, i.e., to lose everything almost surely, can be obtained by approximating prices with a convex combination of traders’ discounted beliefs [12]. In the same framework, using arguments similar to those presented in this paper, one can obtain general conditions to study the long-run dynamics of relative consumption and, eventually, agent survival. When agents’ utility is not time separable, several long-run selection outcomes may occur, included cases in which path dependency emerges [13].

In the present paper, we propose criteria, based on the signs of the differences of relative entropies, that can be applied to generic strategies depending on prevailing market odds. The criteria are simple and abstract from strategy-specific details. It is sufficient to know the odds bettors consider fair and the amount each bettor is willing to bet when the odds are equal to the ones the other bettor believes fair to understand if one bettor will eventually dominate the market or, conversely, if the two bettors will asymptotically retain a finite, and fluctuating, amount of wealth. Interestingly, by considering generic strategies, one recovers the traditional role of luck in the game of chance, often neglected in some of the previously-mentioned studies. Indeed, in our setting, it is generically possible that the ultimate fate of a bettor is not only decided by the adopted strategy, but also by the specific realized sequence of binary events. As examples, we apply the new criteria to the case of Constant Relative Risk Averse (CRRA) bettors and to the case of agents following logit quantal response betting strategies [14,15].

## 2. Model

Consider two agents who make a sequence of consecutive bets against each other on binary events. The rounds of betting are indexed by t∈N, and in each *t*, the outcome of the event $s_{t}\in \left\{0,1\right\}$ is an independent Bernoulli trial with success probability $\pi^{*}$: $s_{t}=1$ means that the event occurs, while $s_{t}=0$ that it does not. In each round *t*, agent i∈{1,2} has to choose the fraction of wealth to be wagered $b_{t}^{i}$ and the side of the bet $\sigma_{t}^{i}\in \left\{0,1\right\}$, where one means betting on the occurrence of the event, while zero betting against it. We assume that the amount bet is redistributed among the winners according to the parimutuel procedure, that is proportionally to how much they have bet, without any house-take. Let $p_{t}$ be the prevailing inverse odds ratio at round *t* for the occurrence of the event. Thus, if $s_{t}=1$, the agent betting on the occurrence of the event receives $1/p_{t}$ times the amount bet, while if $s_{t}=0$, the agent betting against the occurrence of the event receives $1/(1-p_{t})$ times the amount bet. Agents’ betting strategies are based on prevailing odds, and they try to maximize their gain by increasing their bet when they perceive favorable opportunities. Following [10], we assume the following:for each agent *i*, there exists a constant “fair” inverse odds ${\bar{p}}^{i}\in \left(0,1\right)$;each agent *i* chooses the side of the bet comparing prevailing odds with those she/he believes fair: she/he bets on the occurrence of the event ($\sigma_{t}^{i}=1$) if the odds are higher than those she/he believes fair ($p_{t}<{\bar{p}}^{i}$), while she/he bets against the occurrence of the event ($\sigma_{t}^{i}=0$) if the odds are lower than those she/he believes fair ($p_{t}>{\bar{p}}^{i}$);for each agent *i*, there exists a continuous betting function $b^{i}\in \left[0,1\right)$ such that:
(a)$b^{i}\left({\bar{p}}^{i}\right)=0$: agent *i* is willing to bet nothing when she/he considers prevailing odds fair,(b)$0<b^{i}\left(p_{t}\right)<1$ when $p_{t}\neq {\bar{p}}^{i}$: agent *i* cannot bet more than what she/he owns. The possibility that she/he bets all her wealth is ruled out as it would lead to wealth zero almost surely.

Without loss of generality we set ${\bar{p}}^{1}<{\bar{p}}^{2}$. Thus, if $w_{t-1}^{i}$ is the wealth of agent i∈{1,2} before the event at time *t* is realized, the prevailing inverse odds $p_{t}$ are set by the equation:(1)$$\frac{w_{t-1}^{1}b^{1}\left(p_{t}\right)}{1-p_{t}}=\frac{w_{t-1}^{2}b^{2}\left(p_{t}\right)}{p_{t}}$$
being always $\sigma_{t}^{1}=0$ and $\sigma_{t}^{2}=1$. We require that the functions $b^{i}$ are such that (Equation 1) admits one and only one solution. This is for instance the case if they are monotonic, strictly concave, or strictly convex on the set of attainable prices. The amount of wealth that is not bet is invested in a risk-less asset that pays no interest. Hence, after the event at round *t* is realized, the wealth of agents is updated according to $w_{t}^{i}=R_{s_{t}}^{i}\left(p_{t}\right)\;w_{t-1}^{i}$ with:(2)$$R_{s}^{i}\left(p\right)=\left(1-b^{i}\left(p\right)\right)+\delta_{s,i-1}\;b^{i}\left(p\right)\;\left(\frac{\delta_{i,1}}{1-p}+\frac{\delta_{i,2}}{p}\right)$$
where $\delta_{a,b}$ is the Kronecker delta and $R_{s}^{i}\left(p\right)$ represents the gross return of wealth of agent *i*, conditional or prevailing odds *p*, and realized outcome *s*. Since the house takes no fee, the aggregate wealth is constant, and we set $w_{t}=w_{t}^{1}+w_{t}^{2}=1$ such that $p_{t}\in \left[{\bar{p}}^{1},{\bar{p}}^{2}\right]$ and $p_{t}={\bar{p}}^{i}$ if and only if $w_{t}^{i}=1$.

## 3. Long-Run Selection

The dynamics of wealth described by (Equation 2) can lead to two different outcomes: either a single agent accrues all the wealth and dominates the market or both agents indefinitely survive, each with a positive, and fluctuating, fraction of wealth. In general, the fate of an agent could depend on the specific sequence of realizations of the random variable $s_{t}$. The behavior of the system can be described using the logarithm of the relative wealth of Agent 2 with respect to Agent 1, $z_{t}=\log\left(w_{t}^{2}/w_{t}^{1}\right)$. Indeed, the asymptotic behavior of $z_{t}$ summarizes all the relevant information about the agents’ fate: $z_{t}$ diverges toward +∞ if and only if $w_{t}^{1}\to 0$ and $w_{t}^{2}\to 1$; $z_{t}$ diverges toward −∞ if and only if $w_{t}^{1}\to 1$ and $w_{t}^{2}\to 0$; $z_{t}$ does not diverge if and only if both agents maintain a strictly positive wealth share in the long-run.

Consider the conditional odds-adjusted wealth growth rate for agent *i* obtained by dividing the realized wealth growth rate by the odds of the realized outcome:$$Q_{0}^{i}\left(p\right)=R_{0}^{i}\left(p\right)\;\left(1-p\right)\;,\;Q_{1}^{i}\left(p\right)=R_{1}^{i}\left(p\right)\;p\;.$$

Due to individual budget constraints and market clearing condition (Equation 1), one has $\sum_{s=0}^{1}Q_{s}^{i}\left(p\right)=1$. Those quantities can be thought of as the wealth shares the agent allocates to the possible realizations of the Bernoulli trial [13,16]. Since in every *t*, each agent *i* keeps a fraction $1-b^{i}\left(p_{t}\right)$ of her/his wealth in the risk-less security, this is *as if* she/he is constantly allocating a share $\left(1-b^{i}\left(p_{t}\right)\right)p_{t}$ of her/his wealth on the realization of the event and a share $\left(1-b^{i}\left(p_{t}\right)\right)\left(1-p_{t}\right)$ against the realization of the event. To those fractions, Agent 1 adds $b^{1}\left(p_{t}\right)$ against the event, while Agent 2 adds $b^{2}\left(p_{t}\right)$ on the occurrence of the event. Then, we define the relative entropy of the odds-adjusted growth rate with respect to the true probability of the outcome:(3)$$I_{\pi^{*}}^{i}\left(p\right)=\pi^{*}\;\log\left(\frac{\pi^{*}}{Q_{1}^{i}\left(p\right)}\right)+\left(1-\pi^{*}\right)\;\log\left(\frac{1-\pi^{*}}{Q_{0}^{i}\left(p\right)}\right)\;.$$

Under the wealth shares interpretation of $\left\{Q_{s}^{i}\left(p\right)\right\}$, $I_{\pi^{*}}^{i}\left(p\right)$ can be interpreted as a measure of how different from the best possible allocation agent *i*’s one is [11]. We know from the literature [3,4,5,10] that the best allocation is the Kelly rule, which corresponds to allocating wealth to each possible realization of the Bernoulli trial according to the true probabilities, $\pi^{*}$ and $1-\pi^{*}$.

It is immediate to verify that the drift of the process $z_{t}$ conditional on prevailing odds is just the difference of the relative entropy (Equation 3) of the odds-adjusted growth rates of the two agents:(4)$$\mu_{z}\left(p\right)=E\left[z_{t}-z_{t-1}|p\right]=I_{\pi^{*}}^{1}\left(p\right)-I_{\pi^{*}}^{2}\left(p\right)\;.$$

The agent who gains wealth in expectation is the agent whose odds-adjusted growth rates have the lowest relative entropy with respect to the true probabilities at prevailing odds, or, equivalently, the one whose wealth shares allocated to the realizations of the Bernoulli trial have the lowest relative entropy with respect to the Kelly rule at prevailing odds. Studying the details of the trajectory of the agents’ relative wealth would require detailed knowledge of the betting strategies. Nonetheless, in order to characterize the long-run behavior of the model, such full knowledge is not necessary.

**Proposition** **1.**
*Let $\sigma =\{s_{1},s_{2},\ldots \}$ denote a realization of the Bernoulli process, and let $w_{t}^{i}\left(\sigma \right)$ be the associated sequence of agent i’s wealth. If agents’ betting strategies satisfy the requirements of Section 2, then:*
(i)
*if $I_{\pi^{*}}^{1}\left({\bar{p}}^{1}\right)>I_{\pi^{*}}^{2}\left({\bar{p}}^{1}\right)$ and $I_{\pi^{*}}^{1}\left({\bar{p}}^{2}\right)>I_{\pi^{*}}^{2}\left({\bar{p}}^{2}\right)$, then almost surely Agent 2 dominates: she/he accrues all the wealth, $\lim_{t\to \infty }w_{t}^{2}=1$ and $\lim_{t\to \infty }p_{t}={\bar{p}}^{2}$;*
(ii)
*if $I_{\pi^{*}}^{1}\left({\bar{p}}^{1}\right)<I_{\pi^{*}}^{2}\left({\bar{p}}^{1}\right)$ and $I_{\pi^{*}}^{1}\left({\bar{p}}^{2}\right)<I_{\pi^{*}}^{2}\left({\bar{p}}^{2}\right)$, then almost surely Agent 1 dominates: she/he accrues all the wealth, $\lim_{t\to \infty }w_{t}^{1}=1$ and $\lim_{t\to \infty }p_{t}={\bar{p}}^{1}$;*
(iii)
*if $I_{\pi^{*}}^{1}\left({\bar{p}}^{2}\right)<I_{\pi^{*}}^{2}\left({\bar{p}}^{2}\right)$ and $I_{\pi^{*}}^{1}\left({\bar{p}}^{1}\right)>I_{\pi^{*}}^{2}\left({\bar{p}}^{1}\right)$, then almost surely both agents survive: they retain a positive amount of wealth, ${lim sup}_{t\to \infty }w_{t}^{i}>0$ for i=1,2, and prevailing odds fluctuate in $\left({\bar{p}}^{1},{\bar{p}}^{2}\right)$;*
(iv)
*if $I_{\pi^{*}}^{1}\left({\bar{p}}^{2}\right)>I_{\pi^{*}}^{2}\left({\bar{p}}^{2}\right)$ and $I_{\pi^{*}}^{1}\left({\bar{p}}^{1}\right)<I_{\pi^{*}}^{2}\left({\bar{p}}^{1}\right)$, then either $\lim_{t\to \infty }w_{t}^{2}\left(\sigma \right)=1$ or $\lim_{t\to \infty }w_{t}^{1}\left(\sigma \right)=1$ depending on the realization of the Bernoulli process σ.*



**Proof.** Define the (conditional) increment $g\left(p,s\right)=z_{t+1}-z_{t}$ when $p_{t}=p$ and $s_{t+1}=s$. From (Equation 2), remembering that, by hypothesis, $b^{1}\left(p\right)$ and $b^{2}\left(p\right)$ cannot be both zero for the same *p* and are continuous, it is immediate to see that:
$$\log\frac{1-B^{2}}{1+B^{1}{\bar{p}}^{2}/\left(1-{\bar{p}}^{2}\right)}<g\left(p,0\right)<0<g\left(p,1\right)<\log\frac{1+B^{2}\left(1-{\bar{p}}^{1}\right)/{\bar{p}}^{1}}{1-B^{1}}\;,$$
where $B^{i}=\max\left\{b^{i}\left(x\right)|{\bar{p}}^{1}\leq x\leq {\bar{p}}^{2}\right\}$. Thus, the increments *g* are finite and bounded, and Theorems 2.2, 3.1, and 3.2 of [17] can be applied to the process $\left\{z_{t}\right\}$. Notice that $\lim_{z\to -\infty }E\left[z_{t+1}-z_{t}|z_{t}=z\right]=\mu_{z}\left({\bar{p}}^{1}\right)$ and $\lim_{z\to +\infty }E\left[z_{t+1}-z_{t}|z_{t}=z\right]=\mu_{z}\left({\bar{p}}^{2}\right)$. Thus, if $\mu_{z}\left({\bar{p}}^{1}\right),\mu_{z}\left({\bar{p}}^{2}\right)>0$, then $\lim_{t\to \infty }z_{t}=+\infty$, whence (i). If $\mu_{z}\left({\bar{p}}^{1}\right),\mu_{z}\left({\bar{p}}^{2}\right)<0$, then $\lim_{t\to \infty }z_{t}=-\infty$, whence (ii). If $\mu_{z}\left({\bar{p}}^{1}\right)>0$ and $\mu_{z}\left({\bar{p}}^{2}\right)<0$, then there exists a finite interval *A* such that $z_{t}\in A$ almost surely for any *t*, whence (iii). If $\mu_{z}\left({\bar{p}}^{1}\right)<0$ and $\mu_{z}\left({\bar{p}}^{2}\right)>0$, then on any Bernoulli sequence, either $\lim_{t\to \infty }z_{t}=+\infty$ or $\lim_{t\to \infty }z_{t}=-\infty$, whence (iv). □

In order to decide the survival or dominance of agents, it is not generically necessary to know all the details of the investment strategies, but simply the Bernoulli probability $\pi^{*}$, the inverse odds considered fair by the two agents, ${\bar{p}}^{1}$ and ${\bar{p}}^{2}$, and two positive numbers, $b^{1}\left({\bar{p}}^{2}\right)$ and $b^{2}\left({\bar{p}}^{1}\right)$, representing the fraction of wealth one agent bets if the odds are equal to those the other agent would consider fair. These quantities are sufficient to compute the relative entropy $I_{\pi^{*}}^{i}\left({\bar{p}}^{j}\right)$ with i,j∈{1,2} that appears in Proposition 1. An informationally-constrained external observer, who knows the true probabilities driving the occurrence of events, but does not have perfect knowledge of individual behaviors, can thoroughly infer long-run selection outcomes exploiting only a very limited amount of information about the agents’ strategies [14,15]. A similar result, based on similar techniques, can be obtained in a general equilibrium model with complete markets, provided agent preferences satisfy certain assumptions [13].

It is immediate to see that if $\pi^{*}>{\bar{p}}^{2}$, we are in Case (i), while if $\pi^{*}<{\bar{p}}^{1}$, we are in Case (ii), recovering a result in [10]. The definitions of survival and dominance in [10] are weaker than the ones adopted here. Given the relative simplicity of the considered process, however, their conclusions are still valid. Notice that the dominance of any agent in Case (iv) is not only realized on peculiar zero-measure sequences, like the sequence of all ones or the sequence of all zeros, but on sets of sequences with finite probability. This is where luck enters into the picture: both agents might dominate and accrue all wealth, but only Fortuna will decide who.

## 4. Example with CRRA Bettors

The betting strategies introduced in Section 2 are flexible enough to accommodate several behavioral prescriptions. As an illustrative example, in this section, we consider the case in which agents bet to maximize the expected utility of wealth using a power utility function with Constant Relative Risk Aversion (CRRA). Call $\gamma^{i}>0$ the relative risk aversion coefficient of agent *i* and $\pi^{i}$ the subjective probability (belief) that agent *i* assigns to the realization of the event, which is precisely the inverse odds that agent *i* would consider fair. Assuming $\pi^{1}<\pi^{2}$, for $p_{t}\in \left[\pi^{1},\pi^{2}\right]$, Agent 1 bets against the occurrence of the event a fraction of wealth $b^{1}$ that maximizes $\pi^{1}{\left(1-b^{1}\right)}^{1-\gamma^{1}}+\left(1-\pi^{1}\right){\left(1-b^{1}p_{t}/\left(1-p_{t}\right)\right)}^{1-\gamma^{1}}$ to obtain:(5)$$b^{1}\left(p_{t}\right)=\frac{{\left(p_{t}\left(1-\pi^{1}\right)\right)}^{\frac{1}{\gamma^{1}}}-{\left(\pi^{1}\left(1-p_{t}\right)\right)}^{\frac{1}{\gamma^{1}}}}{{\left(p_{t}\left(1-\pi^{1}\right)\right)}^{\frac{1}{\gamma^{1}}}+p_{t}{\left(\pi^{1}\right)}^{\frac{1}{\gamma^{1}}}{\left(1-p_{t}\right)}^{\frac{1-\gamma^{1}}{\gamma^{1}}}}\;.$$

Conversely, Agent 2 bets in favor of the realization of the event a fraction of wealth $b^{2}$ that maximizes $\pi^{2}{\left(1+b^{2}\left(1-p_{t}\right)/p_{t}\right)}^{1-\gamma^{2}}+\left(1-\pi^{2}\right){\left(1-b^{2}\right)}^{1-\gamma^{2}}$ to obtain:(6)$$b^{2}\left(p_{t}\right)=\frac{{\left(\pi^{2}\left(1-p_{t}\right)\right)}^{\frac{1}{\gamma^{2}}}-{\left(p_{t}(1-\pi^{2}\right)}^{\frac{1}{\gamma^{2}}}}{{\left(\pi^{2}\left(1-p_{t}\right)\right)}^{\frac{1}{\gamma^{2}}}+\left(1-p_{t}\right){\left(1-\pi^{2}\right)}^{\frac{1}{\gamma^{2}}}{\left(p_{t}\right)}^{\frac{1-\gamma^{2}}{\gamma^{2}}}}\;.$$

The positive risk aversion implies that agents never bet the totality of their wealth. Figure 1 provides two examples of how agents’ betting strategies vary depending on the inverse odds ratio. In the effective price support, betting strategies are always continuous and strictly concave, so that the equilibrium market odds always exist and are unique. If we set $\gamma^{i}=1$ for i=1,2, we recover the case of Kelly betting: agents maximize the expected log-growth rate of their wealth. In this case, previous contributions [3,4,5,10] showed that the agent whose beliefs had a lower relative entropy with respect to the truth dominates in the long-run. In the other cases, instead, the selection dynamics are richer. Figure 2 reports the long-run selection outcomes inferred using the conditions from Proposition 1. Depending on agents’ risk aversion and beliefs, any case of Proposition 1 may generically occur. Notice how low risk aversion and asymmetric beliefs enhance the role of luck in deciding the ultimate winner. This is in line with previous findings about path-dependent, long-run selection outcomes [11,13].

## 5. Example with Logit Quantal Response Bettors

In this section, we use our criteria to assess long-run selection outcomes of a repeated betting market where agents’ behavior is described by a quantal response function [14,15]. Suppose that agent *i* wants to bet a fraction of wealth $x_{0}^{i}$ against the realization of the event and a fraction $x_{1}^{i}$ on the realization of the event. As in the previous example, the agent assigns a subjective probability $\pi^{i}$ to the realization of the event. Her/his expected payoff for round *t* is:$$U^{i}\left(x_{0}^{i},x_{1}^{i}\right)=\pi^{i}\left(\left(1-x_{0}^{i}\right)+1+x_{1}^{i}\frac{1-p_{t}}{p_{t}}\right)+\left(1-\pi^{i}\right)\left(1+x_{0}^{i}\frac{p_{t}}{1-p_{t}}+\left(1-x_{1}^{i}\right)\right)\;.$$

The agent maximizes the expected payoff under the constraints $x_{0}^{i}+x_{1}^{i}=1$ and $I\left(x_{0}^{i},x_{1}^{i}\right)\geq H^{i}$, where $I(x_{0}^{i},x_{1}^{i})$ is the entropy of the portfolio $\left(x_{0}^{i},x_{1}^{i}\right)$ and $H^{i}$ is the minimum level of portfolio entropy, that is the maximum level of information attainable by the agent. Notice that if $H^{i}=0$, full information, the solution is either $x_{0}^{i}=1,\;x_{1}^{i}=0$ or $x_{0}^{i}=0,\;x_{1}^{i}=1$ depending on $\pi^{i}$ and $p_{t}$, that is a boundary solution. The problem becomes:$$\max_{x_{0}^{i},x_{1}^{i}}\;L=U^{i}\left(x_{0}^{i},x_{1}^{i}\right)+\lambda^{i}\left(-H^{i}-x_{0}^{i}\log x_{0}^{i}-x_{1}^{i}\log x_{1}^{i}\right)+\eta^{i}\left(1-x_{0}^{i}-x_{1}^{i}\right)$$
and given the constraints, we obtain the solutions:$$x_{0}^{i}=\frac{e^{\beta^{i}\frac{p_{t}-\pi^{i}}{1-p_{t}}}}{e^{\beta^{i}\frac{p_{t}-\pi^{i}}{1-p_{t}}}+e^{\beta^{i}\frac{\pi^{i}-p_{t}}{p_{t}}}}\;,\;\;x_{1}^{i}=\frac{e^{\beta^{i}\frac{\pi^{i}-p_{t}}{p_{t}}}}{e^{\beta^{i}\frac{p_{t}-\pi^{i}}{1-p_{t}}}+e^{\beta^{i}\frac{\pi^{i}-p_{t}}{p_{t}}}}\;,$$
where $\beta^{i}>0$ is a monotonic decreasing function of $H^{i}$ and $\beta^{i}\to +\infty$ when $H^{i}\to 0$. The more the agent is informationally constrained, the smaller the β. Notice that agent investment shares $\left(x_{0}^{i},x_{1}^{i}\right)$ are equivalent to those derived under a multinomial random utility model [18]. If $p_{t}>\pi^{i}$, then the agent takes a net position against the event, hence $\sigma_{t}^{i}=0$ and $b^{i}\left(p_{t}\right)=x_{0}^{i}\left(p_{t}\right)-x_{1}^{i}\left(p_{t}\right)>0$. If $p_{t}<\pi^{i}$, then the agent takes a net position in favor of the event, hence $\sigma_{t}^{i}=1$ and $b^{i}\left(p_{t}\right)=x_{1}^{i}\left(p_{t}\right)-x_{0}^{i}\left(p_{t}\right)>0$. If $p_{t}=\pi^{i}$, then the agent takes a risk-less position and $b^{i}\left(p_{t}\right)=0$. It follows that ${\bar{p}}^{i}=\pi^{i}$, as in the previous example. Moreover, it can be easily shown that the betting functions as defined above are monotonic.

Figure 3 shows how two examples of logit quantal response betting functions vary with respect to *p*. Figure 4 reports the long-run selection outcomes inferred using the conditions of Proposition 1. As one can notice, low values of $\beta^{1}$ and $\beta^{2}$, associated with agents with strong informational constraints, allow cases to emerge in which both agents survive, while the occurrence of path-dependent scenarios becomes very likely when the β’s are large.

## 6. Conclusions

In this paper, we considered a market for bets where two agents repeatedly wage on an uncertain event with two possible outcomes using generic betting strategies. We proposed simple criteria, based on the difference of relative entropies, to decide the asymptotic amount of wealth of the two bettors. Little information about agents’ behavior was needed to apply the criteria: it was sufficient to know traders’ fair odds and the amount they were willing to bet at the fair odds of the opponent. Thus, systemic behavior can be inferred also when the model is underdetermined. When generic betting strategies are considered, three outcomes are possible in the long-run: (1) one bettor accrues all the wealth with probability one; (2) both bettors survive with a positive and fluctuating amount of wealth; or (3) one of the two bettors eventually accrues all the wealth with finite probability. In the third case, luck recovers the role of ultimate arbiter, traditionally attributed to it in games of chance. Notice that if one confines the analysis to specific families of strategies, like Kelly or fractional Kelly strategies, the third outcome becomes non-generic or disappears [8,9]. This explains why it was largely unobserved in several previous studies.

## Figures and Tables

**Figure 1 entropy-21-00585-f001:**
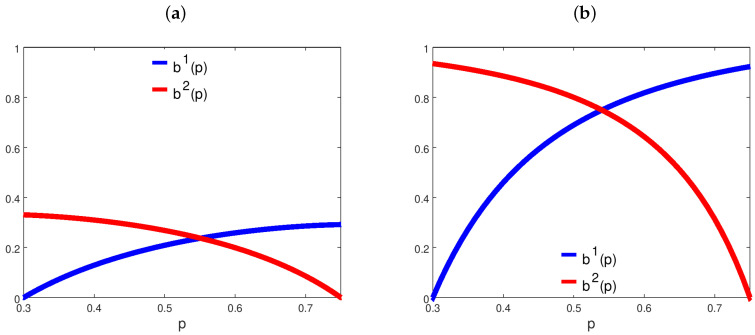
Constant Relative Risk Averse (CRRA) agents’ betting strategies. In both panels, we set ${\bar{p}}^{1}=0.3$ and ${\bar{p}}^{2}=0.75$; (**a**) $\gamma^{1}=\gamma^{2}=2$; (**b**) $\gamma^{1}=\gamma^{2}=0.5$.

**Figure 2 entropy-21-00585-f002:**
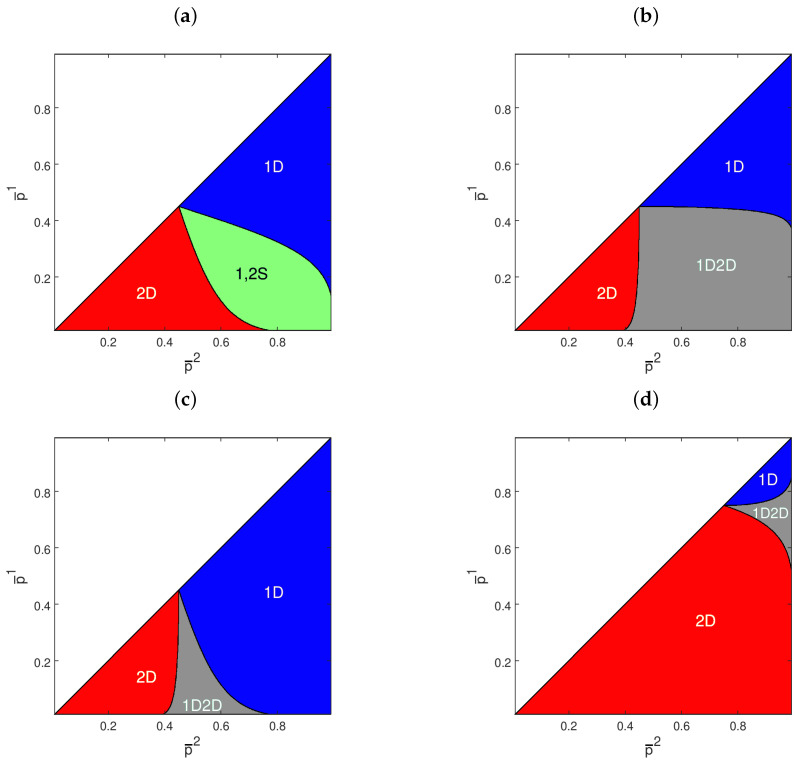
Dominance, survival, and vanishing of CRRA agents for different combinations of fair inverse odd ratios ${\bar{p}}^{i}$. 1D: Agent 1 dominates; 2D: Agent 2 dominates; 1,2S: both agents survive; 1D2D: either Agent 1 or Agent 2 dominates. (**a**) $\pi^{*}=0.45$, $\gamma^{1}=\gamma^{2}=2$; (**b**) $\pi^{*}=0.45$, $\gamma^{1}=\gamma^{2}=0.5$; (**c**) $\pi^{*}=0.45$, $\gamma^{1}=2$, $\gamma^{2}=0.5$; (**d**) $\pi^{*}=0.75$, $\gamma^{1}=0.5$, $\gamma^{2}=2$.

**Figure 3 entropy-21-00585-f003:**
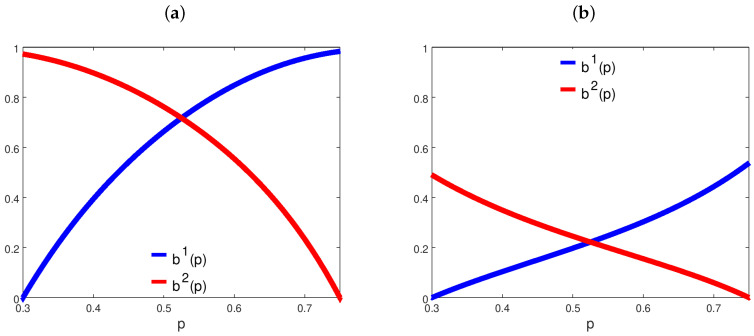
Logit quantal response agents’ betting strategies. In both panels, we set ${\bar{p}}^{1}=0.3$ and ${\bar{p}}^{2}=0.75$; (**a**) $\beta^{1}=\beta^{2}=2$; (**b**) $\beta^{1}=\beta^{2}=0.5$.

**Figure 4 entropy-21-00585-f004:**
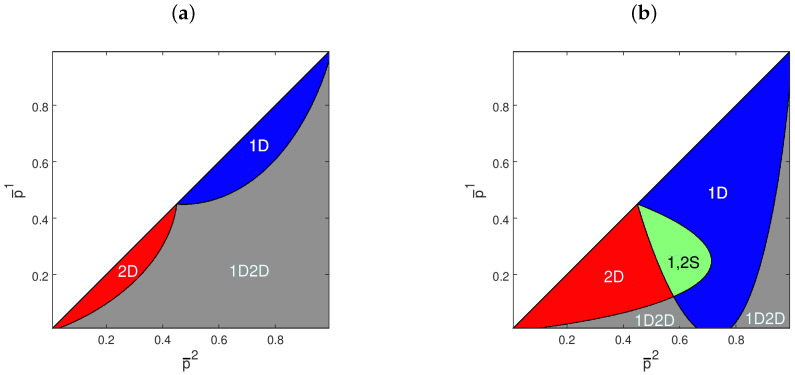
Dominance, survival, and vanishing of logit quantal response agents for different combinations of fair inverse odds ratios ${\bar{p}}^{i}$. 1D: Agent 1 dominates; 2D: Agent 2 dominates; 1,2S: both agents survive; 1D2D: either Agent 1 or Agent 2 dominates. (**a**) $\pi^{*}=0.45$, $\beta^{1}=\beta^{2}=2$; (**b**) $\pi^{*}=0.45$, $\beta^{1}=\beta^{2}=0.5$; (**c**) $\pi^{*}=0.45$, $\beta^{1}=2$, $\beta^{2}=0.5$; (**d**) $\pi^{*}=0.75$, $\beta^{1}=0.5$, $\beta^{2}=2$.

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
