# Peer review of "Betting, Selection, and Luck: A Long-Run Analysis of Repeated Betting Markets"

_entropy, 2019, doi:10.3390/e21060585_

Round 1

Reviewer 1 Report

The MS considers the impact of a relative entropy criteria on a repeated betting market for two agents facing a binary outcome. As the authors correctly point out, the conventional Kelly rule for betting with perfect information is a special case of a more general betting market where agents may not bet according to the Kelly rule, or more importantly, may face informational constraints. In my mind the important contribution contained in the MS is the ability of the model to predict the long-run system level behavior without detailed knowledge of the individual betting strategies. This model seems to be another example of how to infer systemic behavior when the model is underdetermined. In betting markets, the outcome is determined by the actions of individuals, but the detailed betting strategies are generally unknown. The paper by Scharfenaker and Foley (2017) published in this journal may be of some interest to the authors in this respect.

My main concern with the MS is section 3. The authors’ choice of z as representing the behavior of the system needs more justification. The way it reads z seems to be defined in order to justify eq. 4, making the relative entropy claim particular to that description of the system.  I wonder if there is a way to use the principle of maximum entropy to arrive at the same conclusions by specifying appropriate constraints on the system?

The finding that the agent with lower relative entropy (less uncertainty relative to the true probabilities) gains wealth in expectation is not very surprising. But what exactly does it mean when an agent’s odd adjusted growth rate has a lower relative entropy and what are the situational implications for relative entropy differences? Here is an interesting parallel to Scharfenaker and Foley (2017) and Foley (2017) who constraint individual agents’ mixed strategies to have a minimum informational entropy. Without perfect information agents respond to lotteries with logit-quantal response probabilities parameterized by a “decision temperature” that generates endogenous fluctuations in the responses. The decision temperature represents individual agents’ informational limitations and when two agents have two different decision temperatures, one is typically able to realize a net economic surplus at the expense of the other. But, as observers of a system, we also face informational constraints, for example on the detailed betting strategies of individuals. Informationally constrained agents and informationally constrained observers turn out to be dual to one another.

In the introduction the authors initially discuss the implications of agents that do not bet according to the Kelly rule or who are not perfectly informed. Is there a connection between the relative entropy of the odd-adjusted growth rate and a minimum informational entropy constraint for the agents? If so, this should be made clearer.

It seems as if the authors approach is more along the lines of informationally constrained observers to a betting market than to informationally constrained agents, though again, these are dual problems. Still it would be nice if this distinction was addressed at some point.

References

Ellis Scharfenaker and Duncan K. Foley, 2017. "Quantal Response Statistical Equilibrium in Economic Interactions: Theory and Estimation," Entropy, 19, 444; doi:10.3390/e19090444.

Duncan K. Foley, 2017. "Information theory and behavior," Working Papers 1731, New School for Social Research, Department of Economics.

Author Response

Thank you very much for taking the time of reading our paper. We really appreciate your feedback. We took the liberty of offering in our conclusions the general interpretation of the result you provided in your report.

1) Concerning the use of the variable z, we clarified its origin and why it is natural to use it to describe the dynamics of the system. It was admittedly given too much for granted in the previous version. The use of MEP to derive the dynamical evolution of the system is outside our reach (for now at least). It is interesting that the referee points it out because this is precisely one of the directions in which we would like to extend our work in the future. More precisely, we would like to derive a sort of "Lyapunov function" that might be used to describe the long-run statistical equilibrium of the system, as entropy does for instance in thermodynamics. Sadly, such a result escaped us so far.

2) We agree that odd adjusted growth rates were not sufficiently explained. We added a comment to clarify their meaning.

3) The papers by Scharfenaker and Foley (2017) and Foley (2017) are definitely of interest. Thank you for pointing them out. We added a new example to our paper directly inspired by their contributions. This goes in the direction of clarifying the connection between the relative entropy of the odds-adjusted growth rate and the minimum informational entropy constraint for the agents. Probably more can be done in this direction, but not inside the time and space limits of this revision.

4) The need for distinguishing between "informationally constrained observers" and "informationally constrained agents" is a very good point. We comment on it in the new version of the paper.

Please also notice that in order to follow the suggestions of other referees, the title of the paper has changed.

Reviewer 2 Report

The paper itself is potentially interesting, but the quality of the paper should be improved before its acceptance for publication.

More specifically:

1) The title is somewhat misleading, like "New Results" will not stay new in a long run. I would suggest something along the lines of "Limiting Behavior of Wealth (of Agents) in Repeated Betting Market". It can be something else if the authors come up with a better version.

2) I would suggest to include author's names in front of the numerical reference (I think that can be done automatically by using \citet instead of \cite in LaTeX) for clarity of the presentation.

3) The outline of the model is confusing for non-expert readers. In contrast, the cited reference Kets et al. [8] seems to be much clearer and I would suggest to reach their level of clarity. This can be done by itemizing the assumptions in the model setup. It is easy to miss the fact that the betting takes place consecutively and the agents use fixed non-adaptive odds.

4) Figure 1 should be improved, it should be plotted in color and with more pleasant looking fonts. Same for Figure 2 - choose better font.

5) This is perhaps the most critical point. It is not clear why the choice of parameter gamma = 1 corresponding to the log utility(?) is not discussed (including formulas, Figures, etc.). I would expect that the log utility is the basic canonical example.

Author Response

Thank you very much for taking the time of reading our paper. We really appreciate your feedback. We reply to every single point below.

1) Very good point indeed! "New Results" has been removed from the tile and it has been modified to reflect your suggestion. If it still concerns you, let us know.

2) This is something that is decided by the bibliographic style of the journal. I'm afraid it is outside our control. We would prefer to have names too. We tried to rephrase the sentences with numerical citations to be more readable.

3) We followed your advice and shaped the description of the model along the lines of the provided reference. Itemization included.

4) Indeed. It was presumed to be in color but for some reason, it turned out B&W. Now it's in color. Thank you for pointing it out.

5) The case gamma=1 is basically the only case discussed in previous literature. This is why we skip it. This is explicitly mentioned in the revised version of the paper.

Please also notice that in order to follow the suggestions of other referees, the paper now contains one additional example of application.

Reviewer 3 Report

Bottazzi-Giachini manuscript studies the long-run performance of generic betting strategies in a ''market environment'' in which agents with different betting strategies are competing and the prevailing odds, as well as the betting strategies, depend on the past performance of the various agents.

They extend the seminal results on log-optimal portfolios introduced by Kelly (1955) and Cover-Thomas (1992), in which is demonstrated that the rule that ensures the optimal growth rate of capital for a gambler is to bet shares of its capital in proportion equals the true probability of the outcomes. This rule (aka Kelly rule) is the rule that would be optimally chosen by an agent with log utility and correct beliefs. If we call ``effective beliefs" the proportion of  the wealth that  an agent invest in each state, we have the general result that the average log-growth rate of an agent wealth is determined by the difference between the average relative Entropy (K-L divergence) of its effective beliefs and the average relative Entropy of the market odds.

The Economics and Finance literature have made use of these results to study the validity of the market selection hypothesis (Friedman) that agents with incorrect beliefs eventually lose their wealth to agents with correct beliefs, irrespective of agents preferences. 

In general equilibrium models with dynamically complete markets and exogenous beliefs, Sandroni (2000) and Blume-Easley (2006) demonstrate that for a general class of utility functions, agents survival only depends on the accuracy of agents beliefs, even if the investment rule under non-log utility may and typically does differ from the log optimal one. 

The reason is that in this setting preferences affect portfolio allocations and agents optimal saving decision in such a way that the difference in agents saving rates (``effective discount factors'') compensate exactly for the difference between ``effective beliefs'' and beliefs.

Bottazzi-Giachini results are derived in a temporal equilibrium model.

Unlike the General equilibrium setting,  in a temporal equilibrium the saving channel is shoot down and we have again that the log-optimal strategy, with the correct beliefs, is the one that guarantees the fastest growth rate of capital and thus survival. However, when agents utility functions differ from log, the ``effective beliefs'' differ from the agent beliefs and the portfolio strategies endogenously vary over time because they depend on the prevailing market odds, which are path dependent.

This dependence makes it cumbersome to characterize the consumption shares dynamics.  More generally,  the same difficulty occurs when we allow for betting strategies that depend on the prevailing odds in a way that is non constrained to be the solution of a utility maximization problem.

With Proposition 1, the authors present a powerful shortcut. They show that to characterize qualitative the long-run dynamic of the system  (i.e., which agent survives) it suffices to know the relative accuracy of the effective beliefs (betting strategies) of the two agents, calculated when one of the two agents (A,B) dominates. 

The intuition goes as follow.

If agent A (B) betting strategy is more accurate than agent B's (A's) when A(B) dominates and when B(A) dominates,  then the stochastic process of agents consumption shares has a unique stable point (agent A (B) dominates) and it converges to it almost surely.

If agent A(B) betting strategy is better than B's (A's) when A (B) dominates but worse when B(A) dominates, then the stochastic process of agents consumption shares has two local attractors which occur with positive probability. In this case, luck determines which one of the agents dominates. 

If agent A(B) betting strategy is more accurate than B's (A's) when B (A) dominates but less accurate than B's (A's) when A(B) dominates, then the stochastic process of agents consumption shares has two reflecting points and both agents survive with consumption shares that never find a resting point.

These conditions are a special case of the one identified by Dindo (2019).

Dindo (2019) characterize the long-run survival of agents trading in general equilibrium models by decomposing their investment strategies into an ``effective belief'' and an ``effective discount factor'' component.  When analyzing an economy with two agents, agents survival is driven by the difference between their effective saving rates and the relative accuracy (measured in terms of K-L divergence) of their effective beliefs. These two components might depend on agents beliefs and preferences and he presents examples in which agents have non-time-separable preferences that could not be previously analyzed.

Proposition 1 of Bottazzi-Giachini manuscript is a special case of Dindo 2019 condition (pg. 13). The two conditions coincide when Dindo 2019 condition is applied to settings in which the two agents have the same effective saving rate and survival only depends on agents effective beliefs accuracy.

The merit of this manuscript lies in the simplicity of the arguments and examples presented.

REFERENCES

Blume, Lawrence, and David Easley. "If you're so smart, why aren't you rich? Belief selection in complete and incomplete markets." Econometrica 74, no. 4 (2006): 929-966.

Cover, Thomas M., and Joy A. Thomas. Elements of information theory. John Wiley & Sons, 2012.

Dindo, Pietro. "Survival in speculative markets." Journal of Economic Theory 181 (2019): 1-43.

Kelly Jr, John L. "A new interpretation of information rate." In The Kelly Capital Growth Investment Criterion: Theory and Practice, pp. 25-34. 2011.

Sandroni, Alvaro. "Do markets favor agents able to make accurate predictions?." Econometrica 68, no. 6 (2000): 1303-1341.

Author Response

Thank you very much for taking the time of reading our paper. We really appreciate your feedback. In particular, it is important to stress the relation of our work with Dindo (2019). Both papers apply results from Bottazzi and Dindo (2015), which are derived, in turn, from martingale theory of processes on the line. Moreover, Dindo (2019) proposes, as we do, simple survival/dominance conditions that do not require full knowledge about agent behavior. In the revised version of the manuscript, we stressed these similarities.

Please also notice that in order to follow the suggestions of other referees, the title of the paper has changed. Moreover, the paper now contains one additional example of application.

Bottazzi, G.; Dindo, P. Drift criteria for persistence of discrete stochastic processes on the line. LEM Papers Series 2015/26, Laboratory of Economics and Management (LEM), Sant’Anna School of Advanced Studies, Pisa, Italy, 2015.

Round 2

Reviewer 1 Report

The authors have address all of my concerns and have added some very interested results that definitely strengthen the paper.

Reviewer 2 Report

Accept as it is.

Reviewer 3 Report

na